# Decoupling Weight Regularization from Batch Size for Model Compression

## Abstract

Batch size selection affects the generalization ability of gradient descent, and small batch size is usually preferred. Conventionally, weight regularization is performed for every mini-batch without considering when would be the right time to regularize weights during optimization steps. For model compression, which regularizes weights to follow compressed forms, compression-aware training also performs weight compression for every mini-batch to compute the impact of compression on the loss function. In this paper, we propose a new hyper-parameter called "Non-Regularization period" or NR period during which weights are not regularized (or compressed). We first investigate the influence of NR period on regularization using weight decay and weight random noise insertion. Throughout various experiments, we show that stronger weight regularization demands longer NR period (regardless of batch size) to best utilize regularization effects. From our empirical evidence, we argue that weight regularization for every mini-batch allows small weight decay/noise only and limited regularization effects such that there is a need to search for right NR period and weight regularization strength to enhance model accuracy. Consequently, NR period becomes especially crucial for model compression where strong weight regularization is necessary to increase compression ratio. Using various models, we show that simple weight regularization to comply with compression formats along with long NR period is enough to achieve high compression ratio and model accuracy.

## 1 Introduction

For Deep Neural Networks (DNNs), a common training method updates weights by gradient descent and explicit weight manipulation through regularization. To overcome some practical issues of gradient descent on non-convex optimization problems, there have been several enhancements such as learning rate scheduling and adaptive update schemes using momentum and update history (Ruder, 2016). Optimizing batch size is another way to yield efficient gradient descent. Note that large batch size has the advantage of enhancing parallelism of the training system in order to speed up training, critical for DNN research (Dean et al., 2012). Despite such advantages, small batch size is preferred because it improves generalization associated with flat minima search (Keskar et al., 2016) and other hyper-parameter explorations are more convenient (Masters & Luschi, 2018). Small batch size also affects weight regularization if weight updates for gradient descent and weight regularization are supposed to happen for every mini-batch. For example, for weight decay conducted for every mini-batch, if batch size is modified, then the weight decay factor should also be adjusted accordingly (Loshchilov & Hutter, 2017).

Weight regularization is a process that adds information to the model as a way to avoid overfitting (Goodfellow et al., 2016; van Laarhoven, 2017). In this paper, we explore weight compression as a form of weight regularization as it severely restricts the search space of weights (i.e., regularized by compression forms). Moreover, model compression shrinks the effective model size, which is an important regularization principle (Goodfellow et al., 2016) (note that improved model accuracy by model compression is reported (Frankle et al., 2019)). Weights are regularized in numerous ways by model compression. For example, each weight can be pruned (e.g., Han et al. (2015)) or quantized (e.g., Guo et al. (2017)) to yield a sparse model representation or to reduce the number of bits to represent each weight. While weight regularization for model compression can simply be performed once after training, compression-aware training can improve model accuracy by reflecting

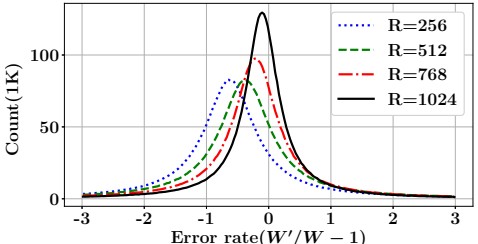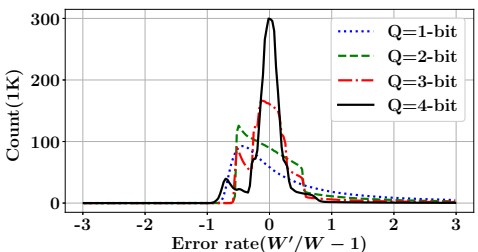

Figure 1: Distribution of weight noise $\epsilon$ after low-rank approximation via SVD (Left) or quantization (Right) when $R$ is the rank and $Q$ is the number of quantization bits.

the impact of model compression on the loss function for every mini-batch update (Courbariaux et al., 2015; Zhu & Gupta, 2017; Zhu et al., 2017; Guo et al., 2017). Now, our question in this paper is the following: *Should weight regularization follow batch size that is considered for gradient descent?* If the answer is *"No,"* then we would need to consider a new hyper-parameter search space with the right weight regularization timing for existing weight regularization methods and model compression schemes.

In this paper, we report the following observations: 1) Model compression tends to induce weight decay and random weight noise; 2) For weight decay and weight noise insertion, less frequent weight regularization allows stronger regularization to best utilize regularization effects regardless of batch size; and 3) Similarly, if weight regularization for compression is performed less frequently, training for model compression permits stronger weight regularization with improved model accuracy, and thus, achieves higher compression ratios.

We confirm our above observations through various experiments and propose an occasional regularization method to decouple weight regularization from batch size selection that is optimized for gradient descent. We verify that our simple model compression techniques (without modifying the underlying training procedures) based on occasional weight regularization (with longer non-regularization period) can achieve higher compression ratio and higher model accuracy compared to previous techniques that demand substantial modifications to the training process.

## 2 NOISE MODEL ON WEIGHT COMPRESSION

We first study the relationship between model compression ratio and weight regularization strength using quantization and singular-value decomposition (SVD) as model compression techniques. We assume a popular quantization method based on binary codes for which a weight vector $\boldsymbol{w}$ is approximated to be $\sum_{i=1}^{q} \alpha_i \boldsymbol{b}_i$ for q-bit quantization, where $\alpha$ is a scaling factor and $\boldsymbol{b}(= \{-1, +1\}^n)$ is a binary vector, and $n$ is the vector size. The quantization error $||\boldsymbol{w} - \sum_i \alpha_i \boldsymbol{b}_i||^2$ is minimized by a method proposed by Xu et al. (2018) to compute $\alpha$ and $\boldsymbol{b}$. For SVD, a weight matrix $\boldsymbol{W} \in \mathbb{R}^{m \times n}$ is approximated to be $\boldsymbol{W}' \in \mathbb{R}^{m \times n}$ by minimizing $||\boldsymbol{W} - \boldsymbol{W}'||$ subject to rank$(\boldsymbol{W}') \leq R$, where $R$ is the target rank.

For our experiments, we use a synthetic ($2048 \times 2048$) weight matrix where each element is randomly generated from the Gaussian distribution $\mathcal{N}(\mu = 0, \sigma^2 = 1)$. Then, we are interested in the amount of change of each weight after quantization and SVD. Assuming that weight noise through compression is expressed as $\epsilon$ in the form of $w' = w(1 + \epsilon)$, Figure 1 shows the distribution of $\epsilon$ with various quantization bits or target ranks. From $\epsilon$ distributions skewed to be negative, it is clear that weights tend to decay more with higher compression ratio, along with a wider range of random noise. Reasonable explanations of Figure 1 would include: 1) weights generated from the Gaussian distribution are uncorrelated such that an approximation step (by compression) using multiple weights would result in noise for each weight, 2) in the case of SVD, elements associated with small eigenvalues are eliminated, 3) averaging effects in quantization reduce the magnitude of large weights. For weight pruning, $\epsilon$ becomes $-1$ or $0$ (i.e., weight decay for selected weights). Correspondingly, we study on weight decay and weight noise insertion in the next two sections as an effort to gain a part of basic knowledge on improved training for model compression, even though actual model compression would demand much more complicated weight noise models.

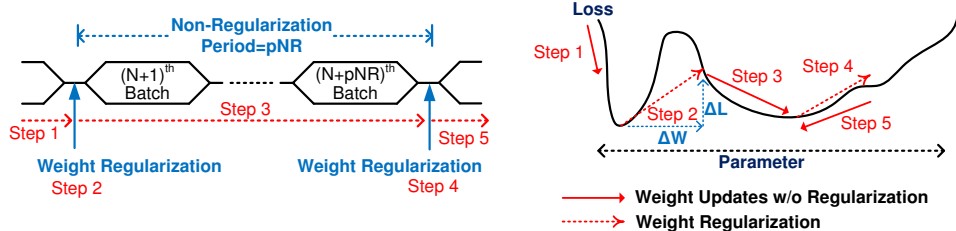

Figure 2: Gradient descent and weight regularization when NR period is given as a multiple of batches. Depending on the loss surface and/or strength of regularization, regularization would lead to step 2 (escaping from a local minimum) or step 5 (returning to a local minimum).

## 3 NON-REGULARIZATION PERIOD

Since weight regularization cannot precede updates for gradient descent, in order to control the frequency of weight regularization, an available option is to skip a few batches without regularization. In this paper, we propose a new hyper-parameter, called "Non-Regularization period" or NR period, to enable occasional regularization and to define the interval of two consecutive regularization events as shown in Figure 2. NR period is expressed as a multiple of batches.

Strong weight regularization facilitates the chance of escaping a local minimum (depicted as step 2 in Figure 2) or require longer NR period to return to a local minimum (described as step 5 in Figure 2). Let us estimate NR period ($pNR$) for a returning case. Given a parameter set $\boldsymbol{w}$ (that is assumed to be close enough to a local minimum) and a learning rate $\gamma$, the loss function of a model $\mathcal{L}(\boldsymbol{w})$ can be approximated as

$$\mathcal{L}(\boldsymbol{w}) \simeq \mathcal{L}(\boldsymbol{w}_0) + (\boldsymbol{w} - \boldsymbol{w}_0)^\top (H(\boldsymbol{w}_0)/2)(\boldsymbol{w} - \boldsymbol{w}_0) \tag{1}$$

using a local quadratic approximation where $H$ is the Hessian of $\mathcal{L}$ and $\boldsymbol{w}_0$ is a set of parameters at a local minimum. After regularization is performed at step $t$, $\boldsymbol{w}$ can be updated by gradient descent as follows:

$$\boldsymbol{w}_{t+1} = \boldsymbol{w}_t - \gamma \frac{\partial \mathcal{L}}{\partial \boldsymbol{w}}\big|_{\boldsymbol{w}=\boldsymbol{w}_t} \simeq \boldsymbol{w}_t - \gamma H(\boldsymbol{w}_0)(\boldsymbol{w}_t - \boldsymbol{w}_0). \tag{2}$$

Thus, after $pNR$, we obtain $\boldsymbol{w}_{t+pNR} = \boldsymbol{w}_0 + (I - \gamma H(\boldsymbol{w}_0))^{pNR} (\boldsymbol{w}_t - \boldsymbol{w}_0)$, where $I$ is an identity matrix. Suppose that $H$ is positive semi-definite and all elements of $I - \gamma H(\boldsymbol{w}_0)$ are less than 1.0, $\boldsymbol{w}_{t+pNR}$ can converge to $\boldsymbol{w}_0$ with long $pNR$ which should be longer with larger $(\boldsymbol{w}_t - \boldsymbol{w}_0)$ (i.e., stronger weight regularization) or smaller $\gamma H(\boldsymbol{w}_0)$. In the next sections, we focus on the relationship between $pNR$ and the strength of weight regularization.

## 4 NR PERIOD STUDY ON WEIGHT DECAY AND WEIGHT NOISE INSERTION

Weight decay is one of the most well-known regularization techniques (Zhang et al., 2018) and different from $L_2$ regularization in a sense that weight decay is separated from the loss function calculation (Loshchilov & Hutter, 2017). Weight decay is performed as

$$\boldsymbol{w}_{t+1} = (1 - \gamma\theta\boldsymbol{w}_t) - \gamma\nabla_{\boldsymbol{w}_t}\mathcal{L}(\boldsymbol{w}), \tag{3}$$

where $\theta$ is a constant weight decay factor. Weight noise insertion is another regularization technique aiminig at reaching flat minima (Goodfellow et al., 2016; Hochreiter & Schmidhuber, 1995). Suppose that random Gaussian noise is added to weights such that $\boldsymbol{w}' = \boldsymbol{w} + \boldsymbol{\epsilon}$ when $\boldsymbol{\epsilon} \sim \mathcal{N}(0, \eta I)$. Then, $\mathcal{L}(\boldsymbol{w}') = \mathbb{E}[f_{\boldsymbol{w}+\boldsymbol{\epsilon}}(\boldsymbol{x}) - \boldsymbol{y}]^2$ where $\boldsymbol{x}$, $\boldsymbol{y}$, $f$ are input, target, and prediction function, respectively. Using Taylor-series expansion to second-order terms, we obtain $f_{\boldsymbol{w}+\boldsymbol{\epsilon}}(\boldsymbol{x}) \approx f_{\boldsymbol{w}}(\boldsymbol{x}) + \boldsymbol{\epsilon}^\top \nabla f(\boldsymbol{x}) + \boldsymbol{\epsilon}^\top \nabla^2 f(\boldsymbol{x})\boldsymbol{\epsilon}/2$. Correspondingly, the loss function can also be approximated as

$$\mathcal{L}(\boldsymbol{w} + \boldsymbol{\epsilon}) \approx \mathbb{E}[f_{\boldsymbol{w}}(\boldsymbol{x}) - \boldsymbol{y}]^2 + \eta\mathbb{E}[(f_{\boldsymbol{w}}(\boldsymbol{x}) - \boldsymbol{y})\nabla^2 f_{\boldsymbol{w}}(\boldsymbol{x})] + \eta\mathbb{E}||\nabla f_{\boldsymbol{w}}(\boldsymbol{x})||^2, \tag{4}$$

where the second term disappears near a local minimum and the third term induces flat minima. Random noise insertion with other distribution models can be explained in a similar fashion (Goodfellow et al., 2016).

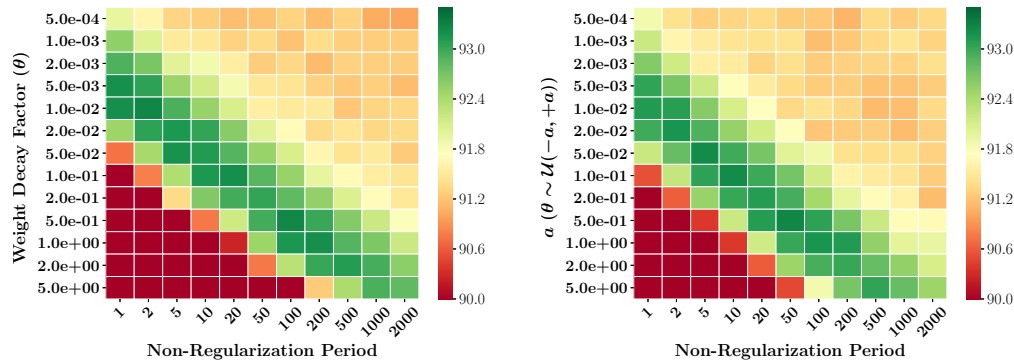

Figure 3: Model accuracy of ResNet-32 on CIFAR-10 using various NR period and amount of weight decay or noise for regularization (original model accuracy without regularization is 92.6%). (Left): Weight decay. (Right): Uniform weight noise insertion.

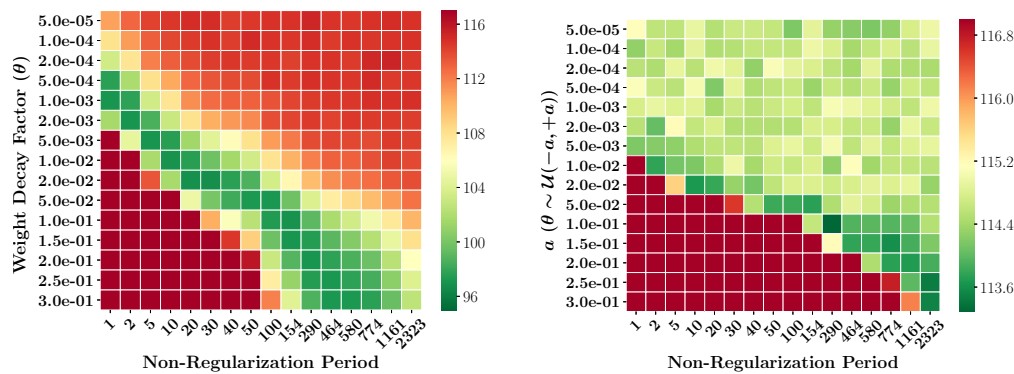

Figure 4: Perplexity of LSTM model on PTB dataset using various NR period and amounts of weight decay or noise. (Left): Weight decay. (Right): Uniform weight noise insertion.

We study the impact of NR period on weight decay and weight noise insertion using ResNet-32 on CIFAR-10 model (He et al., 2016) and a long short-term memory (LSTM) model on PTB dataset (Zaremba et al., 2014). For LSTM model, we use 2 layers with 200 hidden units and the hyper-parameter set introduced by Zaremba et al. (2014). For weight noise model, we plug $\theta \sim \mathcal{U}(-a, +a)$ (uniform distribution) into Eq. (3) to simplify the experiments. Figure 3 shows model accuracy of ResNet-32 given different NR period and weight decay factors. For both weight decay and weight noise insertion, the choice of $\theta$ (representing the amount of each weight regularization) has a clear correlation with NR period (refer to Appendix for training and test accuracy graphs). If we wish to apply stronger weight regularization, then such weight regularization should be conducted less frequently (i.e., larger weight decay factor requires longer NR period) to maximize the regularization effect and achieve high model accuracy. Similar observations are discovered by LSTM model on PTB as shown in Figure 4. Lower perplexity (indicating better generalization) is obtained when NR period increases as weight regularization becomes stronger for each regularization event.

Note that compared with a conventional weight decay factor selection (i.e., $\theta$ in Eq. (3) when $pNR = 1$), weight decay factor can be approximately 1,000 times larger with $pNR \approx 1,000$ in Figure 3 and Figure 4. If $e_w$ is the mean absolute difference given as ($= \mathbb{E}[\|\boldsymbol{w} - \boldsymbol{w}'\|]$), where $\boldsymbol{w}'$ is a weight vector after regularization, then Figure 3 and Figure 4 imply that optimal $e_w/pNR$ seems to be constant. Consequently, optimal $e_w$ depends on $pNR$, and hence, we argue that a wide exploration of various NR period (for occasional regularization) and different weight regularization strength is necessary in order to best utilize regularization effects on a DNN model while previous attempts employ $pNR = 1$ only. In addition, *the observation that stronger weight regularization is enabled by longer $pNR$ is our basic training principle for model compression.*

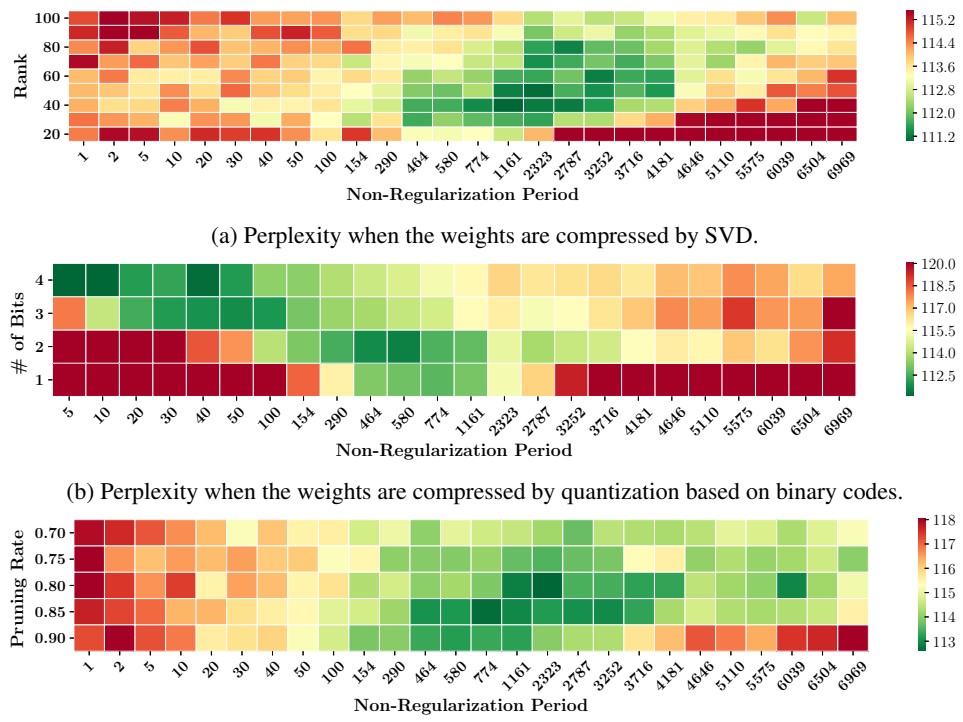

(a) Perplexity when the weights are compressed by SVD.

(b) Perplexity when the weights are compressed by quantization based on binary codes.

(c) Perplexity when the weights are compressed by magnitude-based pruning.

Figure 5: Model accuracy of an LSTM model on PTB compressed by quantization, low-rank approximation or pruning. Original perplexity without model compression is 114.6. For more details, refer to Figure 14.

# 5   NR PERIOD FOR MODEL COMPRESSION

As discussed, weight compression incurs a much more complicated weight regularization model than weight decay or uniform weight noise insertion because 1) as shown in Figure 1, diversified noise models need to be combined to describe weight regularization after model compression and 2) compression-aware training methods would reduce the strength of weight regularization as training is performed with more epochs and weights converge to a compressed form. Nonetheless, we can conjecture that the best training scheme for model compression may require the condition of $pNR \neq 1$ that can be empirically justified.

We apply weight quantization, low-rank approximation (SVD), and pruning to an LSTM model on PTB that we selected for the previous section. We do not modify underlying training principles and use the following simple strategy:

1) Train the model during NR period (as if model compression is not being considered.)
2) Then, perform weight compression in the form of $\boldsymbol{w}' = h(\boldsymbol{w})$.
3) With new full-precision weight $\boldsymbol{w}'$, repeat the above two steps.

$h(\boldsymbol{w})$ can be a magnitude-based pruning (i.e., $h(w)=w$ if $|w|$ is larger than a certain threshold, or $h(w)=0$, otherwise), $\alpha\boldsymbol{b}$ for quantization, SVD function, or even as-yet undiscovered functions.

Figure 5 shows model accuracy associated with a number of different sets of $pNR$ and model compression strength (i.e., target rank for low-rank approximation and the number of quantization bits). Optimal $pNR$ for the best model accuracy is definitely much larger than 1. To explain how Figure 5 is aligned with Section 4, we investigate the relationship between model accuracy and the average of $e_w/pNR$ ($e_w = \mathbb{E}[\|\boldsymbol{w} - \boldsymbol{w}'\|]$, where $\boldsymbol{w}'$ is a weight vector after weight decay, SVD, or pruning) during entire training. We first determine optimal $e_w/pNR$ ($= e_{opt}$) for the best model accuracy. As discussed in Section 4, $e_{opt}$ is assumed to be constant regardless of compression ratio or decay

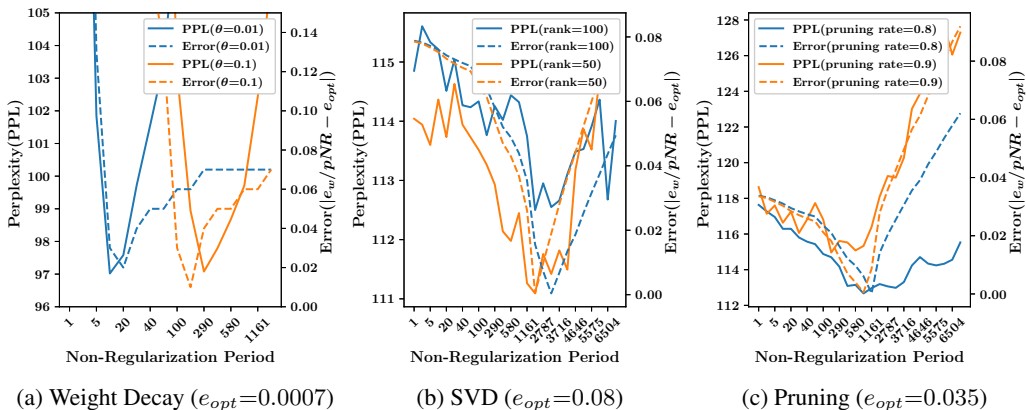

Figure 6: Relationship between model accuracy and error (defined as the difference between $e_w/pNR$ and $e_{opt}$) using PTB LSTM model when weights are regularized by weight decay, SVD, or pruning.

factor, and obtained by finding hyper-parameter sets associated with maximum model accuracy in Figure 4 and Figure 5 and by taking the average of corresponding $e_w$ values. When error is defined to be $|e_w/pNR - e_{opt}|$, Figure 6 shows test perplexity and error of PTB LSTM model with different $pNR$. Such defined error is affected by $pNR$ as shown in Figure 6, and indeed, when error approaches to an optimal value we gain improved model accuracy. Unlike weight decay where $e_w$ is directly computed by decay factors, for model compression techniques, $e_w$ is not directly related to compression-related hyper-parameters (such as ranks and pruning rates). As a result, while Figure 4 shows clear correlation between decay factors and $pNR$ for best model accuracy, Figure 5 suggests that compression ratio and $pNR$ are weakly correlated. Hence, $pNR$ is a hyper-parameter to be determined empirically for model compression. Nonetheless, optimal $pNR$ is definitely much larger than 1, as shown in Figure 6, and decoupled from batch size selection. That means weight regularization for model compression needs to be conducted much less frequently compared with gradient descent since batch size selection considers generalization ability of gradient descent, not regularization effects.

Note that periodic compression has been introduced in the literature to gradually improve compression ratio or automate hyper-parameter search process. DropPruning repeats dropping weights randomly and retraining the model while some previously dropped weights are unpruned until pruning rate reaches a target number (Jia et al., 2018). Weights are incrementally quantized to improve model accuracy (Zhou et al., 2017) or the number of quantization bits can be controlled differently for each layer by a loop based on reinforcement learning (Elthakeb et al., 2018). Structured pruning and fine-tuning process can be iterated to increase pruning rate (Molchanov et al., 2016; Liu et al., 2017). All of these previous works assume $pNR = 1$ (i.e., performing compression for every mini batch) while the goal is increasing compression ratio slowly or finding a set of hyper-parameters through iterative fine-tuning stages. Our proposed compression technique can be combined with such periodic compression methods (incremental compression or automatic hyper-parameter selection are also applicable to our proposed method). In the work by He et al. (2018), soft filter pruning is conducted with $pNR = 1$ epoch without analysis of why such occasional pruning improves model accuracy.

# 6 COMPARISON WITH PREVIOUS MODEL COMPRESSION TECHNIQUES

In this section, we compare some of previous model compression techniques with our compression scheme that introduces $pNR$ and obviates special training algorithm modifications.

## 6.1 FINE-GRAINED WEIGHT PRUNING

The initial attempt of pruning weights was to locate redundant weights by computing the Hessian to calculate the sensitivity of weights to the loss function (LeCun et al., 1990). However, such

Table 1: Pruning rate comparison using LeNet-300-100 and LeNet-5 models on MNIST dataset. DC (Deep Compression) and Sparse VD represent a magnitude-based technique (Han et al., 2016) and variational dropout method (Molchanov et al., 2017), respectively.

| Model | Layer | Weight Size | Pruning Rate (%) | | | |
|---|---|---|---|---|---|---|
| | | | DC | DNS | Sparse VD | Ours |
| LeNet-300-100 | FC1 | 235.2K | 92 | 98.2 | 98.9 | 98.9 |
| | FC2 | 30K | 91 | 98.2 | 97.2 | 96.0 |
| | FC3 | 1K | 74 | 94.5 | 62.0 | 62.0 |
| | Total | 266.2K | 92 | 98.2 | 98.6 | 98.4 |
| LeNet-5 | Conv1 | 0.5K | 34 | 85.8 | 67 | 60.0 |
| | Conv2 | 25K | 88 | 96.9 | 98 | 97.0 |
| | FC1 | 400K | 92 | 99.3 | 99.8 | 99.8 |
| | FC2 | 5K | 81 | 95.7 | 95 | 95.0 |
| | Total | 430.5K | 92 | 99.1 | 99.6 | 99.5 |

a technique has not been considered to be practical due to significant computation overhead for computing the Hessian. Magnitude-based pruning (Han et al., 2015) has become popular because one can quickly find redundant weights by simply measuring the magnitude of weights. Since then, numerous researchers have realized higher compression ratio largely by introducing Bayesian inference modeling of weights accompanying supplementary hyper-parameters.

For example, dynamic network surgery (DNS) (Guo et al., 2016) permits weight splicing when a separately stored full-precision weight becomes larger than a certain threshold. Optimizing splicing threshold values, however, necessitates extensive search space exploration, and thus, longer training time. Variational dropout method (Molchanov et al., 2017) introduces an explicit Bayesian inference model for a prior distribution of weights, which also induces various hyper-parameters and increased computational complexity.

We perform magnitude-based pruning at every $pNR$ step. As a result, even though weights are pruned and replaced with zero at $pNR$ steps, pruned weights are still updated in full precision during NR period. If the amount of updates of a pruned weight grows large enough between two consecutive regularization steps, then the weight pruned at last $pNR$ step may not be pruned at the next $pNR$ step. Such a feature (i.e., pruning decisions are not fixed) is also utilized for weight splicing in DNS (Guo et al., 2016). Weight splicing in DNS relies on a hysteresis function (demanding sophisticated fine-tuning process with associated hyper-parameters) to switch pruning decisions. Pruning decisions through our scheme, on the other hand, are newly determined at every $pNR$ step.

We present experimental results with LeNet-5 and LeNet-300-100 models on MNIST dataset which are also reported by Guo et al. (2016); Molchanov et al. (2017). LeNet-5 consists of 2 convolutional layers and 2 fully connected layers while 3 fully connected layers construct LeNet-300-100. We train both models for 20000 steps using Adam optimizer where batch size is 50. All the layers are pruned at the same time and the pruning rate increases gradually following the equation introduced in Zhu & Gupta (2017):

$$p_t = p_f + (p_i - p_f) \left(1 - \frac{t - t_i}{t_f - t_i}\right)^E, \qquad (5)$$

where $E$ is a constant, $p_f$ is the target pruning rate, $p_i$ is the initial pruning rate, $t$ is the current step, and the pruning starts at training step $t_i$ and reaches $p_f$ at training step $t_f$. After $t_f$ steps, pruning rate is maintained to be $p_f$. For LeNet-5 and LeNet-300-100, $t_i$, $p_i$, $E$ are 8000 (step), 25(%), and 7, respectively. $t_f$ is 12000 (step) for LeNet-5 and 13000 (step) for LeNet-300-100. Note that these choices are not highly sensitive to test accuracy as discussed in Zhu & Gupta (2017). We exclude dropout to improve the accuracy of LeNet-300-100 and LeNet-5 since pruning already works as a regularizer (Han et al., 2015; Wan et al., 2013). We keep the original learning schedule and the total number of training steps (no additional training time for model compression).

Table 1 presents the comparison on pruning rates (see Appendix for test accuracy). Despite the simplicity, our pruning scheme produces higher pruning rate compared with DNS and similar com-

pared with variational dropout technique which involves much higher computational complexity. For Table 1, we use $pNR$=10 for LeNet-5 and $pNR$=5 for LeNet-300-100.

## 6.2 LOW-RANK APPROXIMATION

We apply our proposed occasional regularization algorithm integrated with Tucker decomposition (Tucker, 1966) to convolutional neural network (CNN) models and demonstrate superiority of $pNR$-based scheme over conventional training methods. In CNNs, the convolution operation requires a 4D kernel tensor $\mathcal{K} = \mathbb{R}^{d \times d \times S \times T}$ where each kernel has $d \times d$ dimension, $S$ is the input feature map size, and $T$ is the output feature map size. Then, following the Tucker decomposition algorithm, $\mathcal{K}$ is decomposed into three components as

$$\tilde{\mathcal{K}}_{i,j,s,t} = \sum_{r_s=1}^{R_s} \sum_{r_t=1}^{R_t} \mathcal{C}_{i,j,r_s,r_t} \boldsymbol{P}_{s,r_s}^S \boldsymbol{P}_{t,r_t}^T, \tag{6}$$

where $\mathcal{C}_{i,j,r_s,r_t}$ is the reduced kernel tensor, $R_s$ is the rank for input feature map dimension, $R_t$ is the rank for output feature map dimension, and $\boldsymbol{P}^S$ and $\boldsymbol{P}^T$ are 2D filter matrices to map $\mathcal{C}_{i,j,r_s,r_t}$ to $\mathcal{K}_{i,j,s,t}$. Each component is obtained to minimize the Frobenius norm of ($\tilde{\mathcal{K}}_{i,j,s,t} - \mathcal{K}_{i,j,s,t}$). As a result, one convolution layer is divided into three convolution layers, specifically, $(1 \times 1)$ convolution for $\boldsymbol{P}^S$, $(d \times d)$ convolution for $\mathcal{C}_{i,j,r_s,r_t}$, and $(1 \times 1)$ convolution for $\boldsymbol{P}^T$ (Kim et al., 2016).

In prior tensor decomposition schemes, model training is performed as a fine-tuning procedure after the model is restructured and fixed (Lebedev et al., 2015; Kim et al., 2016). On the other hand, our training algorithm is conducted for Tucker decomposition as follows:

Step 1: Perform normal training for $pNR$ (batches) without considering Tucker decomposition

Step 2: Calculate $\mathcal{C}$, $\boldsymbol{P}^S$, and $\boldsymbol{P}^T$ using Tucker decomposition to obtain $\tilde{\mathcal{K}}$

Step 3: Replace $\mathcal{K}$ with $\tilde{\mathcal{K}}$

Step 4: Go to Step 1 with updated $\mathcal{K}$

After repeating a number of the above steps towards convergence, the entire training process should stop at Step 2, and then the final decomposed structure is extracted for inference. Because the model is not restructured except in the last step, Steps 2 and 3 can be regarded as special steps to encourage wide search space exploration so as to find a compression-friendly local minimum where weight noise by decomposition does not noticeably degrade the loss function.

Using the pre-trained ResNet-32 model with CIFAR-10 dataset (He et al., 2016; Kossaifi et al., 2019), we compare two training methods for Tucker decomposition: 1) typical training with a decomposed model and 2) $pNR$-based training, which maintains the original model structure and occasionally injects weight noise through decomposition. Using an SGD optimizer, both training methods follow the same learning schedule: learning rate is 0.1 for the first 100 epochs, 0.01 for the next 50 epochs, and 0.001 for the last 50 epochs. Except for the first layer, which is much smaller than the other layers, all convolution layers are compressed by Tucker decomposition with rank $R_s$ and $R_t$ selected to be $S$ and $T$ multiplied by a constant number $R_c$ ($0.3 \leq R_c \leq 0.7$ in this experiment). Then, the compression ratio of a convolution layer is $d^2 ST / (SR_s + d^2 R_s R_t + TR_t)$ $= d^2 ST / (S^2 R_c + d^2 R_c^2 ST + T^2 R_c)$, which can be approximated to be $1/R_c^2$ if $S = T$ and $d \gg R_c$. $pNR$ is chosen to be 200.

Figure 7 shows test accuracy after Tucker decomposition[1] by two different training methods. Note that test accuracy results are evaluated only at Step 3 where the training process can stop to generate a decomposed structure. In Figure 7, across a wide range of compression ratios (determined by $R_c$), the proposed scheme yields higher model accuracy compared to typical training. Note that even higher model accuracy than that of the pre-trained model can be achieved by our method if the compression ratio is small enough. In fact, Figure 8 shows that our technique improves training loss and test accuracy throughout the entire training process. Initially, the gap of training loss and test accuracy between pre-regularization and post-regularization is large. Such a gap, however, is quickly reduced through training epochs. Overall, ResNet-32 converges successfully through the

---

[1]https://github.com/larry0123du/Decompose-CNN

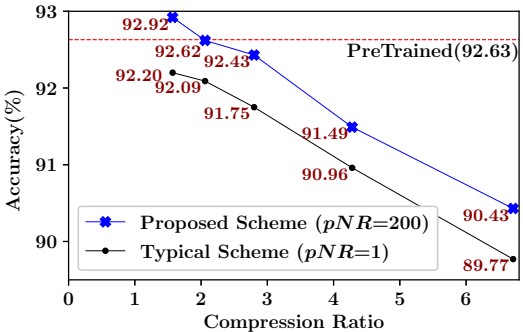

Figure 7: Test accuracy comparison on ResNet-32 using CIFAR-10 trained by typical training method and the proposed training method with various compression ratios. For the proposed scheme, test accuracy is measured only at Step 3 that allows to extract a decomposed structure, and $pNR$ is 200.

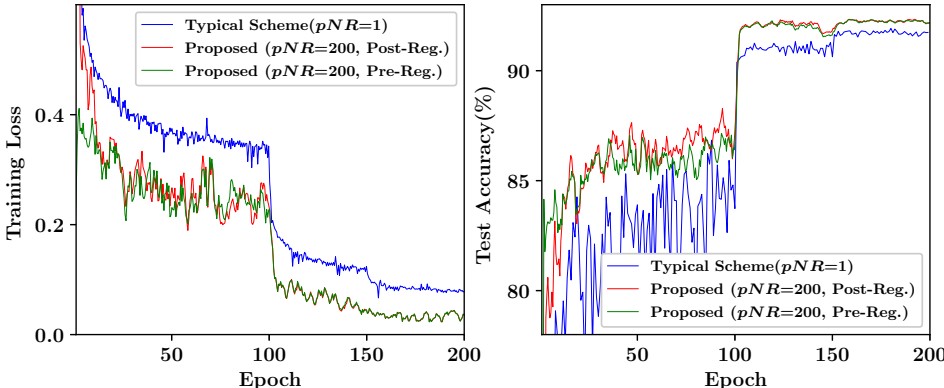

Figure 8: Training loss and test accuracy of ResNet-32 using CIFAR-10. For the proposed scheme, training loss and test accuracy are only monitored right before or after weight regularization for compression (pre-regularization or post-regularization). Compression ratio is 2.8 with $R_c$=0.5.

entire training process with lower training loss and higher test accuracy compared with a typical training method.

For ResNet-34 on ImageNet experiments and VGG19 on CIFAR-10 (including additional compression techniques), refer to Appendix.

## 7 CONCLUSION

In this paper, we show that weight regularization should be decoupled from batch size to enhance regularization effects. To accomplish such decoupling, we introduce a new hyper-parameter called non-regularization period or NR period during which weights are updated only for gradient computations. A larger amount of weight decay and weight noise insertion need to be supported by longer NR period to maximize the regularization effect. We find that model compression with higher compression ratio is also better trained by longer NR period. We demonstrate that various DNN models can be compressed by pruning, quantization, and low-rank approximation successfully with longer NR period while the underlying training algorithm do not need to be modified.

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

# A APPENDIX

## A.1 SUPPLEMENTARY EXPERIMENTS FOR WEIGHT DECAY AND WEIGHT NOISE

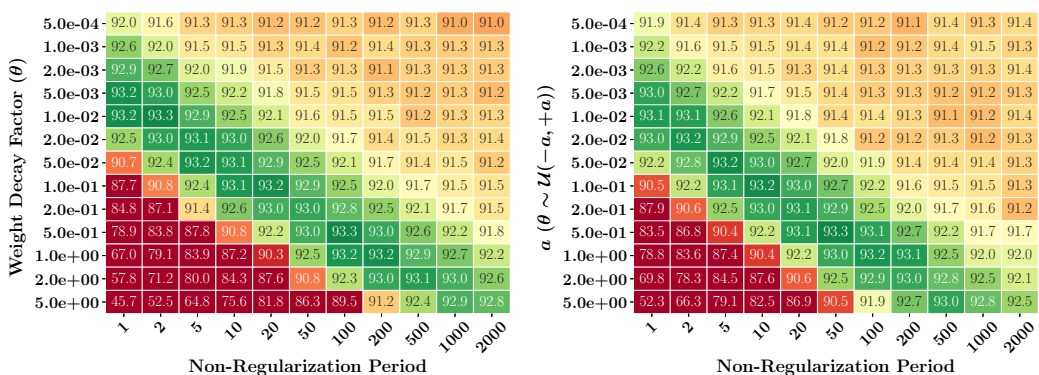

Figure 9: Model accuracy of ResNet-32 on CIFAR-10 using various NR period and amount of weight regularization (original model accuracy without regularization is 92.6%). (Left): Weight decay. (Right): Uniform weight noise insertion.

Table 2: Model accuracy of ResNet-32 on CIFAR-10 and LSTM model on PTB with various weight decay factor and corresponding $pNR$.

| Model | | | Weight Decay Factor($\theta$) | | | | | |
|---|---|---|---|---|---|---|---|---|
| | | 0 | 1e-4 | 5e-4 | 1e-3 | 5e-3 | 1e-2 | 5e-2 |
| ResNet-32 | Accuracy(%) | 92.6 | 93.3 | 93.2 | 93.2 | 93.3 | 93.2 | 92.9 |
| | optimal $pNR$ | N/A | 2 | 5 | 20 | 100 | 200 | 1000 |
| LSTM on PTB | Perplexity | 114.6 | 108.1 | 97.7 | 97.1 | 97.1 | 97.0 | 97.2 |
| | optimal $pNR$ | N/A | 1 | 1 | 1 | 5 | 10 | 100 |

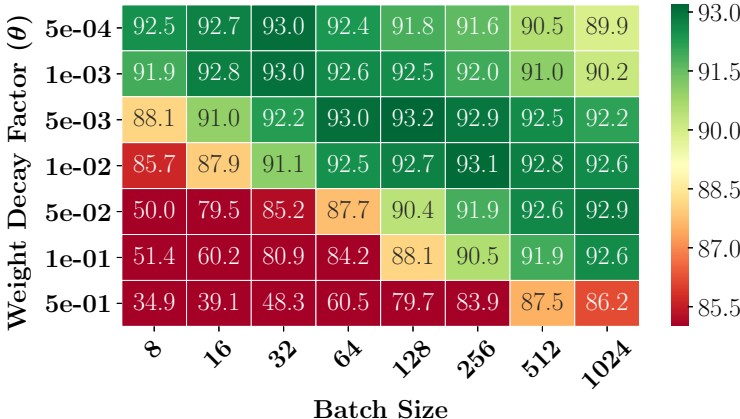

Figure 10: Model accuracy (%) of ResNet-32 for various weight decay factors and batch size when $pNR$=1. Large batch size demands larger weight decay factors that is also reported by Loshchilov & Hutter (2017).

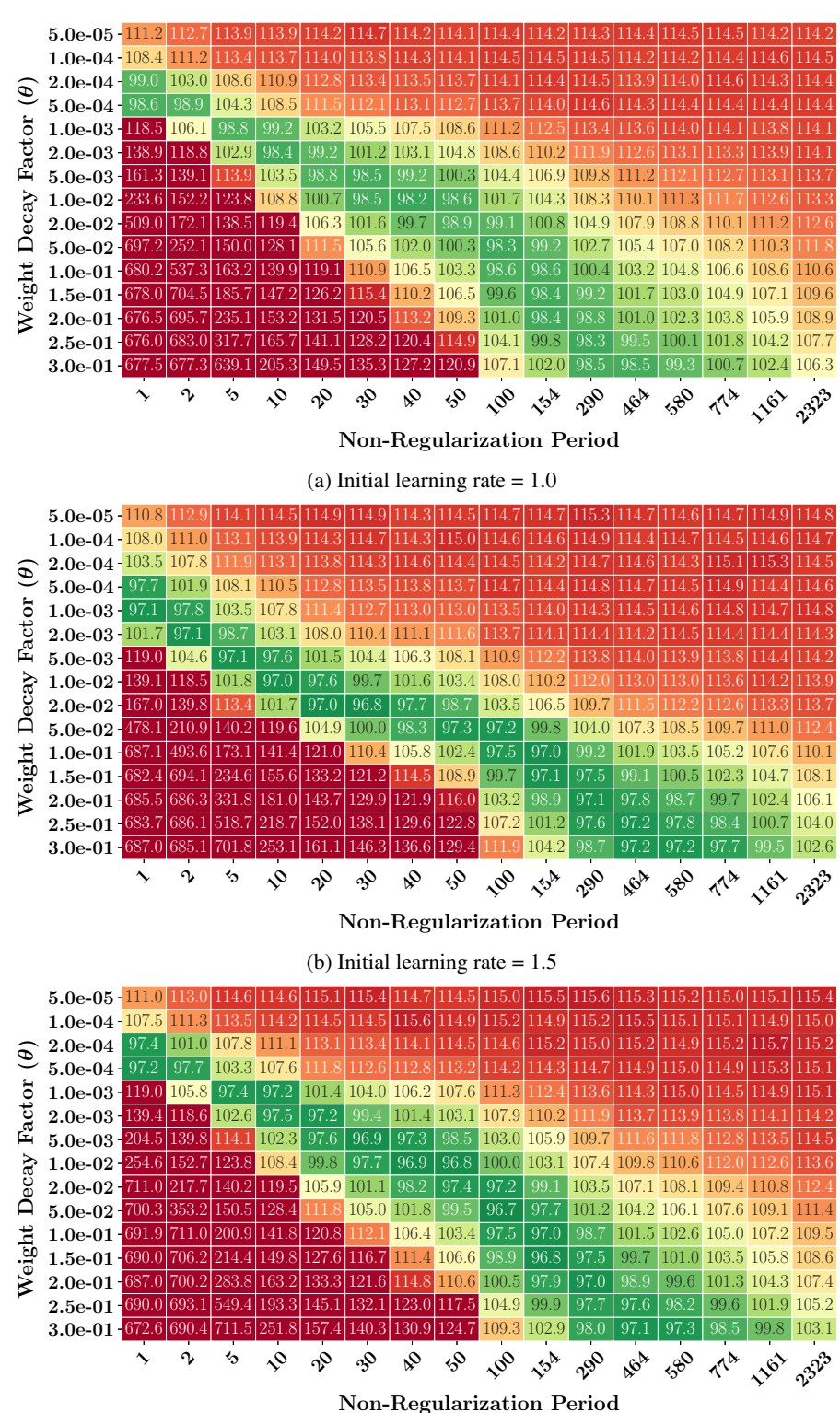

Figure 11: Perplexity of LSTM model on PTB dataset using various NR period and amounts of weight decay (original perplexity without regularization is 114.60).

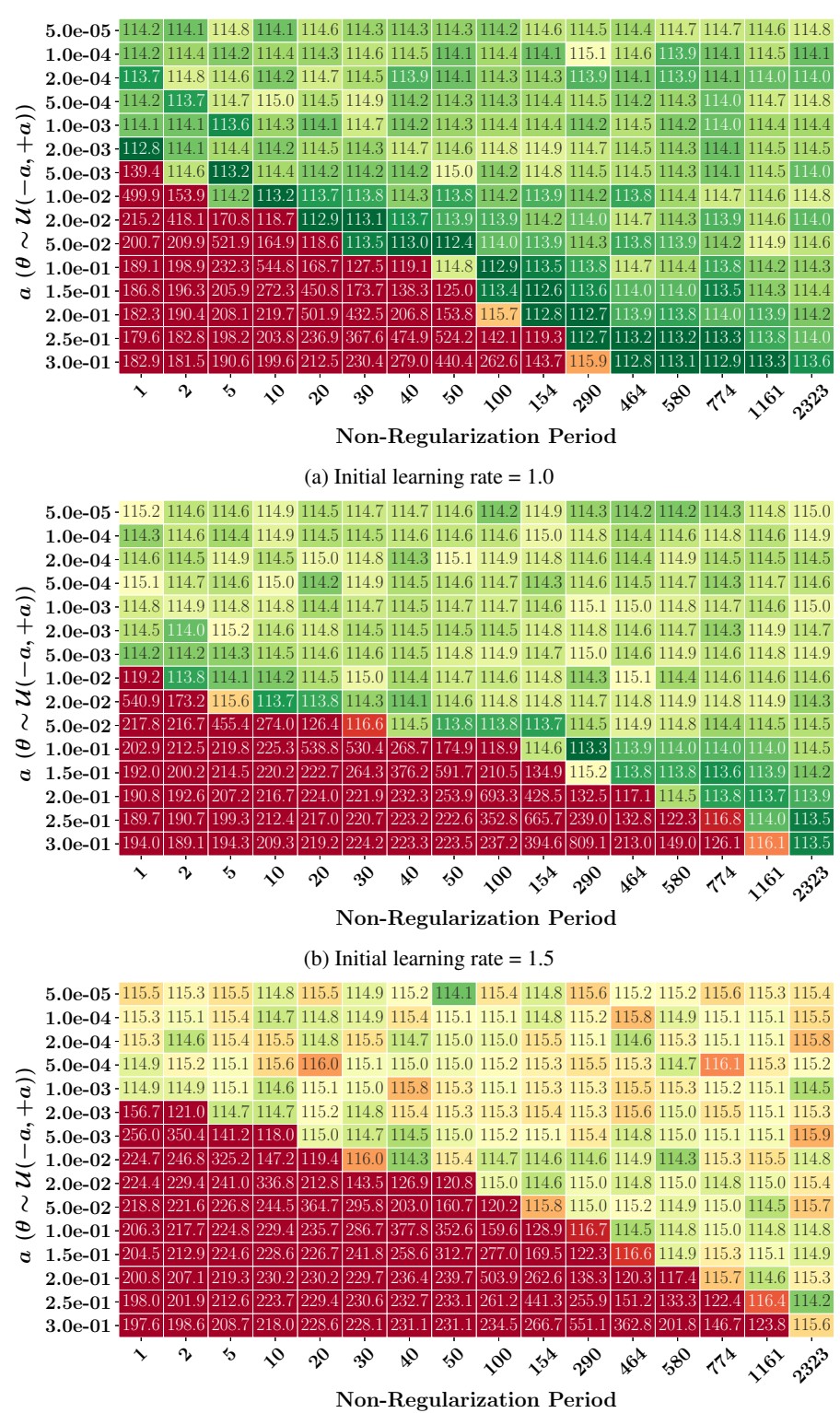

Figure 12: Perplexity of LSTM model on PTB dataset using various NR period and amounts of uniform noise (original perplexity without regularization is 114.60).

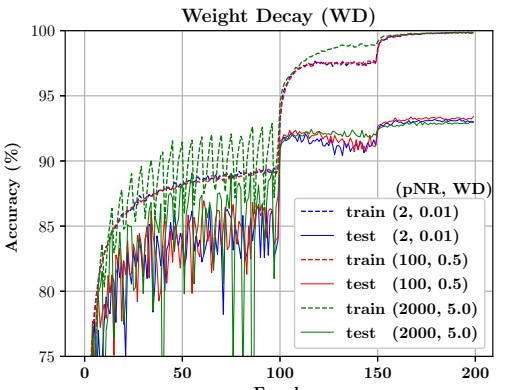 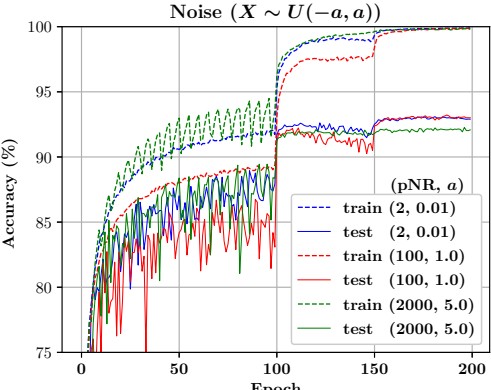

Figure 13: Training and test accuracy of ResNet-32 on CIFAR-10 for variuos NR period and the amount of weight regularization. (Left): Weight decay. (Right): Uniform weight noise.

## A.2 PARAMETER PRUNING

Table 3: Model accuracy comparison using LeNet-300-100 and LeNet-5 models on MNIST dataset. DC (Deep Compression) and Sparse VD represent a magnitude-based technique (Han et al., 2016) and variational dropout method (Molchanov et al., 2017), respectively.

| Model | Accuracy (%) | | | |
|---|---|---|---|---|
| | DC | DNS | Sparse VD | Ours |
| LeNet-300-100 | 98.4 | 98.0 | 98.1 | 98.1 |
| LeNet-5 | 99.2 | 99.1 | 99.2 | 99.1 |

We apply $pNR$-based pruning to an RNN model to verify the effectiveness of $pNR$. We choose an LSTM model (Zaremba et al., 2014) on the PTB dataset (Marcus et al., 1993). Following the model structure given in Zaremba et al. (2014), our model consists of an embedding layer, 2 LSTM layers, and a softmax layer. The number of LSTM units in a layer can be 200, 650, or 1500, depending on the model configurations (referred as small, medium, and large model, respectively). The accuracy is measured by Perplexity Per Word (PPW), denoted simply by perplexity in this paper. We apply gradual pruning with $E = 3$, $t_i = 0$, $p_i = 0$, $t_f = 3^{rd}$ epoch (for medium) or $5^{th}$ epoch (for large) to the pre-trained PTB models. $pNR$-based pruning for the PTB models is performed using $pNR = 100$ and the initial learning rate is 2.0 for the medium model (1.0 for pre-training) and 1.0 for the large model (1.0 for pre-training) while the learning policy remains to be the same as in Zaremba et al. (2014).

Table 4: Comparison on perplexity using various pruning rates. $p_f$ is the target pruning rates for the embedded layer, LSTM layer, and softmax layer.

| Model Size | Pruning Method | Perplexity | | | | | |
|---|---|---|---|---|---|---|---|
| | | $p_f=$ | 0% | 80% | 85% | 90% | 95% | 97.5% |
| Medium | (Zhu & Gupta, 2017) | | 83.37 | 83.87 | 85.17 | 87.86 | 96.30 | 113.6 |
| (19.8M) | Proposed Scheme | | 83.78 | 81.54 | 82.62 | 84.64 | 93.39 | 110.4 |
| Large | (Zhu & Gupta, 2017) | | 78.45 | 77.52 | 78.31 | 80.24 | 87.83 | 103.20 |
| (66M) | Proposed Scheme | | 78.07 | 77.39 | 77.73 | 78.28 | 84.69 | 99.69 |

For all of the pruning rates selected, Table 4 shows that our compression scheme improves perplexity better than the technique in Zhu & Gupta (2017) which is based on Han et al. (2015). The superiority of $pNR$-based pruning is partly supported by the observation that non-zero weights successfully avoid to be small through retraining while the conventional pruning still keeps near-zero (unmasked) weights as depicted in Figure 15.

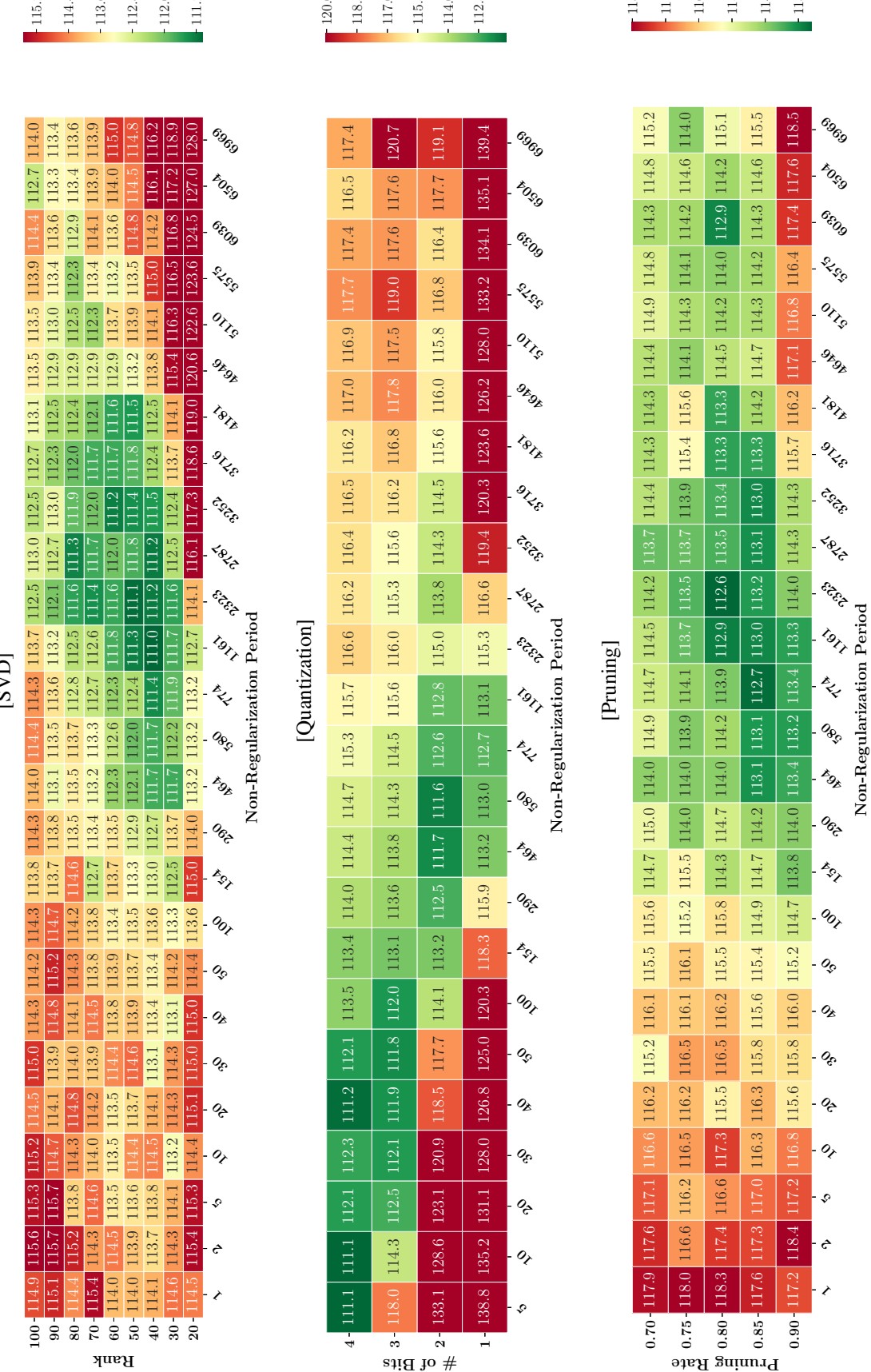

Figure 14: Relationship between test perplexity and *pNR* using PTB LSTM model.

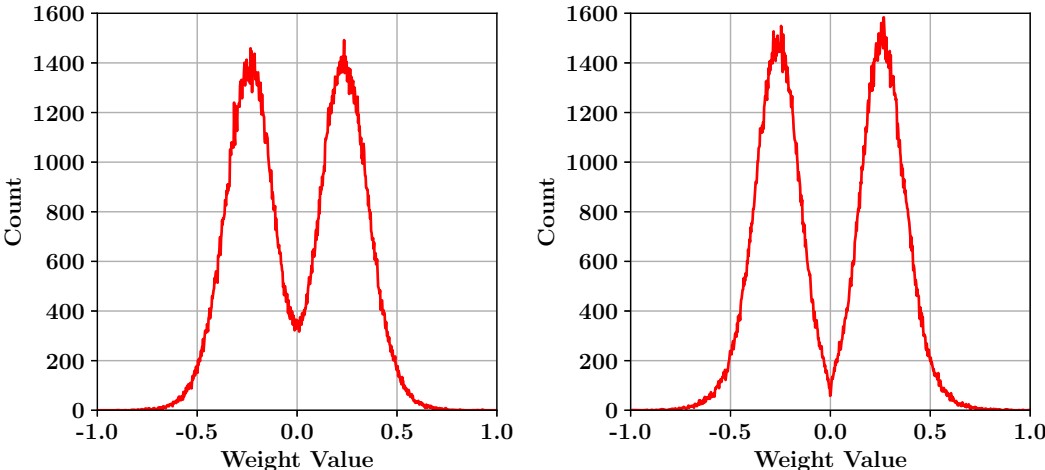

Figure 15: Weight distribution of LSTM layer 1 of the medium PTB model after retraining with (Left) a magnitude-based pruning and (Right) $pNR$-based pruning with 90% pruning rate. Our compression scheme incurs a sharp drop in the count of near-zero weights.

### A.3 TUCKER DECOMPOSITION WITH OCCASIONAL REGULARIZATION

To investigate the effect of NR period on local minima exploration with ResNet-32 on CIFAR-10, Figure 16 presents the changes of loss function and weight magnitude values incurred by occasional regularization. In Figure 16(left), $\Delta\mathcal{L}/\mathcal{L}$ is given as the loss function increase $\Delta\mathcal{L}$ (due to weight regularization at $pNR$ steps) divided by $\mathcal{L}$, which is the loss function value right before weight regularization. In Figure 16(right), $\Delta w$ is defined as $||w - \tilde{w}||_{\mathcal{F}}^2 / N(w)$, where $w$ is the entire set of weights to be compressed, $\tilde{w}$ is the set of weights regularized by Tucker decomposition, $N(w)$ is the number of elements of $w$, and $||X||_{\mathcal{F}}^2$ is the Frobenius norm of $X$. Initially, $w$ fluctuates with large corresponding $\Delta\mathcal{L}$. Then, both $\Delta\mathcal{L}$ and $\Delta w$ decrease and Figure 16 shows that occasional regularization finds flatter local minima (in the view of Tucker decomposition) successfully. When the learning rate is reduced at 100th and 150th epochs, $\Delta\mathcal{L}$ and $\Delta w$ decrease significantly because of a lot reduced local minima exploration space. In other words, occasional regularization helps an optimizer to detect a local minimum where Tucker decomposition does not alter the loss function value noticeably.

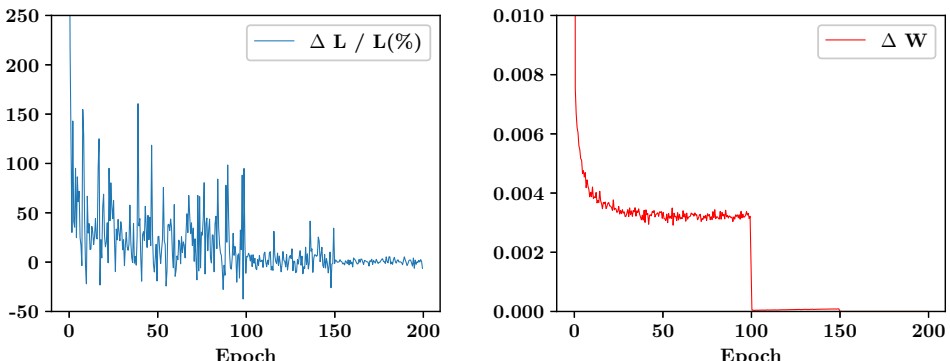

Figure 16: Difference of training loss function and average Frobenius norm of weight values by Step 2 and Step 3 of Figure 2. $R_c = 0.5$ and $pNR = 200$ are used.

### A.4 2-Dimensional SVD Enabled by Occasional Regularization

In this subsection, we discuss why 2D SVD needs to be investigated for CNNs and how occasional regularization enables a training process for 2D SVD.

#### A.4.1 Issues of 2D SVD on Convolution Layers

Convolution can be performed by matrix multiplication if an input matrix is transformed into a Toeplitz matrix with redundancy and a weight kernel is reshaped into a $T \times (S \times d \times d)$ matrix (i.e., a lowered matrix) Chetlur et al. (2014). Then, commodity computing systems (such as CPUs and GPUs) can use libraries such as Basic Linear Algebra Subroutines (BLAS) without dedicated hardware resources for convolution Cho & Brand (2017). Some recently developed DNN accelerators, such as Google's Tensor Processing Unit (TPU) Jouppi et al. (2017), are also focused on matrix multiplication acceleration (usually with reduced precision).

For BLAS-based CNN inference, reshaping a 4D tensor $\mathcal{K}$ and performing SVD is preferred for low-rank approximation rather than relatively inefficient Tucker decomposition followed by a lowering technique. However, a critical problem with SVD (with a lowered matrix) for convolution layers is that two decomposed matrices by SVD do not present corresponding (decomposed) convolution layers, because of intermediate lowering steps. As a result, fine-tuning methods requiring a structurally modified model for training are not available for convolution layers to be compressed by SVD. On the other hand, occasional regularization does not alter the model structure for training. For occasional regularization, SVD can be performed as a way to feed noise into a weight kernel $\mathcal{K}$ for every regularization step. Once training stops at a regularization step, the final weight values can be decomposed by SVD and used for inference with reduced memory footprint and computations. In other words, occasional regularization enables SVD-aware training for CNNs.

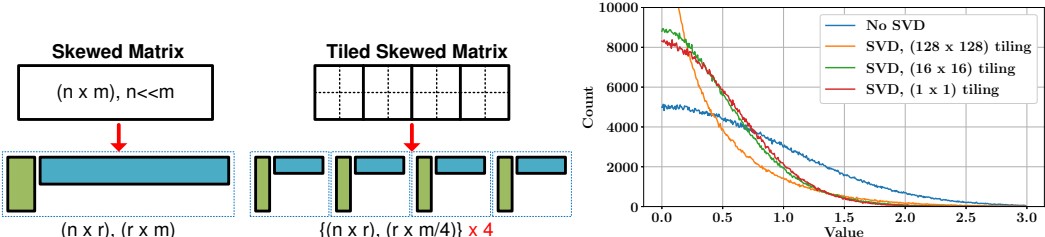

Figure 17: Skewed matrix and a tiling technique are illustrated on the left side, while the right side presents distributions of weights after SVD with different tiling schemes (only positive weights are included).

#### A.4.2 Tiling-Based SVD for Skewed Weight Matrices

A reshaped kernel matrix $\boldsymbol{K} \in \mathbb{R}^{T \times (S \times d \times d)}$ is usually a skewed matrix where row-wise dimension ($n$) is smaller than column-wise dimension ($m$) as shown in Figure 17 (i.e., $n \ll m$). A range of available rank $r$ for SVD, then, is constrained by small $n$ and the compression ratio is approximated to be $n/r$. If such a skewed matrix is divided into four tiles as shown in Figure 17 and the four tiles do not share much common chateracteristics, then tiling-based SVD can be a better approximator and rank $r$ can be further reduced without increasing approximation error. Moreover, fast matrix multiplication is usually implemented by a tiling technique in hardware to improve the weight reuse rate Fatahalian et al. (2004). Hence, tiling could be a natural choice not only for high-quality SVD but also for high-performance hardware operations.

To investigate the impact of tiling on weight distributions after SVD, we tested a $(1024 \times 1024)$ random weight matrix in which elements follow a Gaussian distribution. A weight matrix is divided by $(1 \times 1)$, $(16 \times 16)$, or $(128 \times 128)$ tiles (then, each tile is a submatrix of $(1024 \times 1024)$, $(64 \times 64)$, or $(8 \times 8)$ size). Each tile is compressed by SVD to achieve the same overall compression ratio of $4\times$ for all of the three cases. As described in Figure 17 (on the right side), increasing the number of tiles tends to increase the count of near-zero and large weights (i.e., variance of weight values increases). Figure 17 can be explained by sampling theory where decreasing the number of random samples (of

small tile size) increases the variance of sample mean. In short, tiling affects the variance of weights after SVD (while the impact of such variance on model accuracy should be empirically studied).

Table 5: Test accuracy(%) of ResNet-32 model using CIFAR-10 dataset while the 9 largest convolution layers ($T$=$S$=64, $d$=3) are compressed by SVD using different tiling configurations. For each tile size, rank $r$ is selected to achieve compression ratio of $2\times$ or $4\times$. $pNR$=200 is used for occasional regularization.

| Pre-Trained | Compression Ratio | Size of Each Tile | | | |
|---|---|---|---|---|---|
| | | $64\times64$ | $32\times32$ | $16\times16$ | $8\times8$ |
| 92.63 | $2\times$ | 93.34 ($r$=16) | 93.11 ($r$=8) | 93.01 ($r$=4) | 93.23 ($r$=2) |
| | $4\times$ | 92.94 ($r$=8) | 92.97 ($r$=4) | 93.00 ($r$=2) | 92.81 ($r$=1) |

We applied the tiling technique and SVD to the 9 largest convolution layers of ResNet-32 using the CIFAR-10 dataset. Weights of selected layers are reshaped into $64 \times (64 \times 3 \times 3)$ matrices with the tiling configurations described in Table 5. We perform training with the same learning schedule and $pNR$(=200) used in Section 3. Compared to the test accuracy of the pre-trained model (=92.63%), all of the compressed models in Table 5 achieves higher model accuracy due to the regularization effect of our compression scheme. Note that for each target compression ratio, the relationship between tile size and model accuracy is not clear. Hence, various configurations of tile size need to be explored to enhance model accuracy, even though variation of model accuracy for different tile size is small.

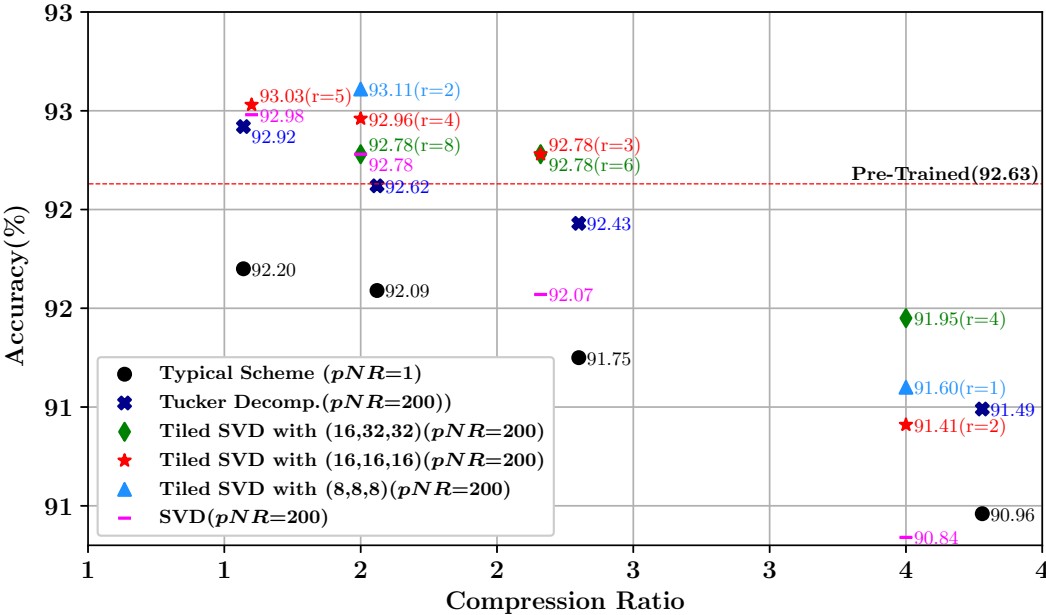

Figure 18: Test accuracy of ResNet-32 model using CIFAR-10 with various target compression ratio and decomposition methods. Except the first small convolution layer, all layers are compressed by the same compression ratio. Convolution layers can be grouped according to 3 different $S$ values (=16, 32, or 64). For tiled SVD, three groups (of different $S$) are tiled in ($k_1 \times k_1$), ($k_2 \times k_2$), or ($k_3 \times k_3$) tile size. ($k_1, k_2, k_3$) configuration is described in legends.

## A.5  Experimental Results on Low-Rank Approximation for CNNs

In this subsection, we apply low-rank approximation trained by occasional regularization to various CNN models.

Figure 18 summarizes the test accuracy values of ResNet-32 (with CIFAR-10 dataset) compressed by various low-rank approximation techniques. Note that tiled SVD and normal SVD are enabled

only by occasional regularization, which obviates model structure modification during training. All configurations in Figure 18 use the same learning rate scheduling and the number of training epochs as described in Section 3. Results show that tiled SVD yields the best test accuracy and test accuracy is not highly sensitive to tile configuration. SVD presents competitive model accuracy for small compression ratios. As compression ratio increases, however, model accuracy using SVD significantly degrades. From Figure 18, tiled SVD associated with occasional regularization is clearly the best low-rank approximation scheme.

Table 6: Comparison on various low-rank approximation schemes of VGG19 (using CIFAR-10 dataset). To focus on convolution layers only, fully-connected layers are compressed by $8\times$ and trained by occasional regularization. Then, fully-connected layers are frozen and convolution layers are compressed (except small layers of $S < 128$) by Tucker decomposition or tiled SVD.

| Comp. Scheme | Parameter | Weight Size | FLOPs | Accuracy(%) |
|---|---|---|---|---|
| Pre-Trained | - | 18.98M | 647.87M | 92.37 |
| Tucker | $R_c$=0.6 | 9.14M (2.08×) | 319.99M (2.02×) | 91.97 |
| Decomposition | $R_c$=0.5 | 6.71M (2.83×) | 235.74M (2.75×) | 91.79 |
| (Typical | $R_c$=0.45 | 5.49M (3.45×) | 191.77M (3.38×) | 91.36 |
| Scheme) | $R_c$=0.4 | 4.61M (4.11×) | 161.60M (4.01×) | 91.11 |
| Tiled | 64×64 ($r$=16) | 9.49M (2.00×) | 316.28M (2.04×) | 92.42 |
| SVD | 64×64 ($r$=11) | 6.52M (2.91×) | 214.25M (3.02×) | 92.33 |
| (Occasional | 64×64 ($r$=10) | 5.93M (3.20×) | 193.85M (3.34×) | 92.23 |
| Regularization, | 64×64 ($r$=9) | 5.55M (3.41×) | 173.44M (3.73×) | 92.22 |
| $pNR$=300) | 64×64 ($r$=8) | 4.74M (4.00×) | 153.04M (4.33×) | 92.07 |

We compare Tucker decomposition trained by a typical fine-tuning process and tiled SVD trained by occasional regularization using the VGG19 model[2] with CIFAR-10. Since this work mainly discusses compression on convolution layers, fully-connected layers of VGG19 are compressed and fixed before compression of convolution layers (refer to Appendix for details on the structure of VGG19). Except for small layers with $S < 128$ (that presents small compression ratio as well), all convolution layers are compressed with the same compression ratio. During 300 epochs to train convolution layers, learning rate is initially 0.01 and is then halved every 50 epochs. In the case of tiled SVD, $pNR$ is 300 and tile size is fixed to be 64×64 (recall that the choice of $pNR$ and tile size do not affect model accuracy significantly). As described in Table 6, while Tucker decomposition with conventional fine-tuning shows degraded model accuracy through various $R_c$, occasional-regularization-assisted tiled SVD presents noticeably higher model accuracy.

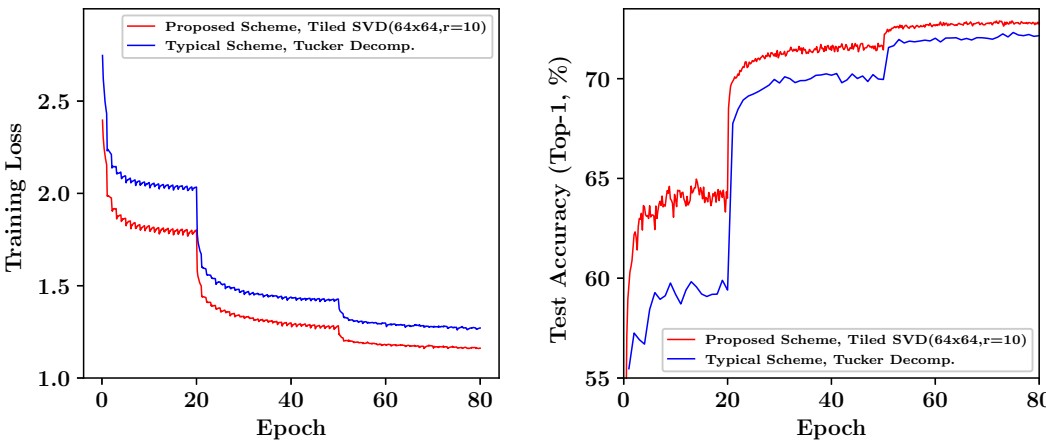

Figure 19: Comparison of two compression schemes on training loss and (top-1) test accuracy of ResNet-34 model using ImageNet. $pNR$=500.

---

[2]https://github.com/chengyangfu/pytorch-vgg-cifar10

We also test our proposed low-rank approximation training technique with the ResNet-34 model[3] He et al. (2016) using the ImageNet dataset Russakovsky et al. (2015). A pre-trained ResNet-34 is fine-tuned for Tucker decomposition (with conventional training) or tiled SVD (with occasional regularization) using the learning rate of 0.01 for the first 20 epochs, 0.001 for the next 30 epochs, and 0.0001 for the remaining 30 epochs. Similar to our previous experiments, the same compression ratio is applied to all layers except the layers with $S < 128$ (such exceptional layers consist of 1.4% of the entire model). In the case of Tucker decomposition, selected convolution layers are compressed with $R_c = 0.46$ to achieve an overall compression of $3.1\times$. For tiled SVD, lowered matrices are tiled and each tile of ($64\times64$) size is decomposed with $r=10$ to match an overall compression of $3.1\times$. As shown in Figure 19, occasional-regularization-based tiled SVD yields better training loss and test accuracy compared to Tucker decomposition with typical training. At the end of the training epoch in Figure 19, tiled SVD and Tucker decomposition achieves 73.00% and 72.31% for top-1 test accuracy, and 91.12% and 90.73% for top-5 test accuracy, while the pre-trained model shows 73.26% (top-1) and 91.24% (top-5).

## A.6 LOWERING TECHNIQUE FOR CNNS

Figure 20 describes a kernel matrix reshaped from a 4D kernel tensor and an input feature map matrix in the form of a Toeplitz matrix. At the cost of redundant memory usage to create a Toeplitz matrix, lowering enables matrix multiplication which can be efficiently implemented by BLAS libraries. A kernal matrix can be decomposed by 2D SVD.

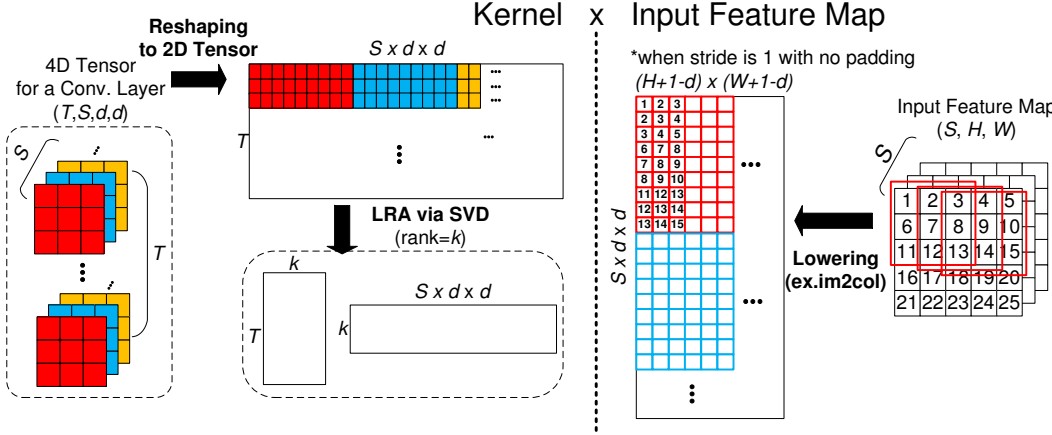

Figure 20: An example of lowering technique using im2col.

## A.7 MODEL DESCRIPTIONS FOR LOW-RANK APPROXIMATION EXPERIMENTS

In this section, we describe model structures and layers selected for low-rank approximation experiments. Small layers close to the input are not compressed because both weight size and compression rate are too small.

---

[3]https://pytorch.org/docs/stable/torchvision/models.html

Table 7: Convolution Layers of ResNet-32 for CIFAR-10

| # of layers | $T$ | $S$ | $d$ | Weight Size | Decomposed |
|---|---|---|---|---|---|
| 1 | 16 | 3 | 3 | 0.4K ( 0.1%) | No |
| 10 | 16 | 16 | 3 | 22.5K ( 5.0%) | Yes |
| 1 | 32 | 16 | 3 | 4.5K ( 1.0%) | Yes |
| 9 | 32 | 32 | 3 | 81.0K (18.0%) | Yes |
| 1 | 64 | 32 | 3 | 18.0K ( 4.0%) | Yes |
| 9 | 64 | 64 | 3 | 324.0K (71.9%) | Yes |
| **Total** | | | | 450.4K (100.0%) | |

Table 8: Convolution and Fully-connected (FC) Layers of VGG-19 for CIFAR-10

| Type | # of layers | $T$ | $S$ | $d$ | Weight Size | Decomposed |
|---|---|---|---|---|---|---|
| | 1 | 64 | 3 | 3 | 0.002M ( 0.01%) | No |
| | 1 | 64 | 64 | 3 | 0.035M ( 0.18%) | No |
| | 1 | 128 | 64 | 3 | 0.070M ( 0.36%) | No |
| Conv. | 1 | 128 | 128 | 3 | 0.141M ( 0.72%) | Yes |
| | 1 | 256 | 128 | 3 | 0.281M ( 1.44%) | Yes |
| | 3 | 256 | 256 | 3 | 1.688M ( 8.61%) | Yes |
| | 1 | 512 | 256 | 3 | 1.125M ( 5.74%) | Yes |
| | 7 | 512 | 512 | 3 | 15.75M (80.37%) | Yes |
| FC | 2 | 512 | 512 | - | 0.500M ( 2.55%) | Yes |
| | 1 | 512 | 10 | - | 0.005M ( 0.02%) | Yes |
| **Total** | | | | | 19.597M (100.0%) | |

Table 9: Convolution Layers of ResNet-34 for ImageNet

| # of layers | $T$ | $S$ | $d$ | Weight Size | Decomposed |
|---|---|---|---|---|---|
| 1 | 64 | 3 | 7 | 0.01M ( 0.04%) | No |
| 6 | 64 | 64 | 3 | 0.21M ( 1.05%) | No |
| 1 | 128 | 64 | 3 | 0.07M ( 0.35%) | No |
| 7 | 128 | 128 | 3 | 0.98M ( 4.90%) | Yes |
| 1 | 256 | 128 | 3 | 0.28M ( 1.40%) | Yes |
| 11 | 256 | 256 | 3 | 6.18M (30.77%) | Yes |
| 1 | 512 | 256 | 3 | 1.13M ( 5.59%) | Yes |
| 5 | 512 | 512 | 3 | 11.25M (55.94%) | Yes |
| Total | | | | 20.11M (100.0%) | |

