# OpenReview forum: "Decoupling Weight Regularization from Batch Size for Model Compression"
_ICLR.cc/2020/Conference — Reject_

### Official Review · AnonReviewer2 · 2019-10-23
**Official Blind Review #2**

**Rating:** 3

**Review:**

This paper studies a model compression algorithm, where training consists in alternatingly updating a full precision weight vector using SGD for a number of steps N, and then compressing the weight vector to a quantized or low-rank representation. The authors present an empirical study of how the properties of this algorithm change as they change the number N.

The contribution is marginal, and the paper is hard to read.
Periodically compressing networks weights, rather than at every iteration, is not new. The paper is missing a "related work" section that embeds the proposed algorithm in the literature.
The experiments are small scale and not exhaustive and there is very little comparison to previous work and alternative methods.

Questions/comments for the authors:
- When you say "regularization" you seem to usually mean "compression"? This was confusing to me.
- What do you mean by "asynchronous regularization"? You seem to mean periodic compression?

**Experience Assessment:**

I have read many papers in this area.

**Review Assessment: Checking Correctness Of Derivations And Theory:**

I assessed the sensibility of the derivations and theory.

**Review Assessment: Checking Correctness Of Experiments:**

I assessed the sensibility of the experiments.

**Review Assessment: Thoroughness In Paper Reading:**

I read the paper at least twice and used my best judgement in assessing the paper.

---

> ### Author Response · Authors · 2019-11-11
> **Response to AnonReviewer2**
>
> We would like to thank you for the review and comments.
> We did our best to improve the manuscript to be read better according to the comments of the reviewers.
> Below we summarized your concerns/questions with our answers.
>
> Q1: Periodic compressing algorithm is not new and "related work" section is missing.
> A1: We tried to find papers with the key words of "periodic compression" and we learned that "periodic compression" may suggest many different techniques. For example, compression-related hyper-parameters can be updated on a regular basis to gradually improve compression ratio, or hyper-parameter search process can be automated by a loop that involves iterative fine-tuning and new hyper-parameter exploration. We added related papers in Section 5 even though such periodic compression techniques are different from our proposed method (most of previous works assume pNR=1). We found one paper introducing soft filter pruning associated with occasional pruning. But there is no any analysis on why such occasional pruning is effective.
> Our proposed technique can be combined with such period compression methods because even though we fixed ranks or quantization bits in this paper (to focus on introducing new training algorithm), such hyper-parameters can be fine-tuned by periodic compression techniques.
>
> Q2: The experiments are small scale and not exhaustive. There is very little comparison to previous work and alternative methods.
> A2: Due to the space limit, some experimental results (including pruning large PTB model and low-rank approximation using ResNet-34 on ImageNet) are described in Appendix. For each compression technique (pruning, quantization, or SVD), we selected well-known models that have been compressed in other papers with high compression ratio. Even though our experimental results could be more exhaustive, we believe that our comparisons using benchmark models are enough to suggest that our proposed compression scheme exhibits competitive (if not better) compression capability while training algorithm for compression can be significantly simplified.
>
> Q3: "Regularization" means "compression"?
> A3: Throughout the manuscript, we tried to deliver the message that model compression is a kind of weight regularization method (i.e., regularization includes compression as a superset). In the revised manuscript, we explicitly use "weight regularization for compression" whenever weights are manipulated to be compressed.
>
> Q4: What do you mean by "asynchronous regularization"? You seem to mean periodic compression?
> A4: We acknowledge that "asynchronous regularization" may be confusing to represent our proposed scheme. On the other hand, "period compression" may be understood in many different ways. We replaced "asynchronous regularization" with "occasional regularization" in the revised manuscript to emphasize that pNR is much larger than 1.

---

> > ### Comment · AnonReviewer2 · 2019-11-11
> > **bumped rating to 3**
> >
> > Thanks for the improvements. I'm (slightly) increasing my rating.

---

> > > ### Author Response · Authors · 2019-11-11
> > > **Thanks a lot! Let us know if we can do better**
> > >
> > > Thank you so much for increasing your rating.
> > > If there are any other ways to increase your rating further, please let us know and we would be very happy to answer your any remaining questions.

---

### Official Review · AnonReviewer1 · 2019-10-23
**Official Blind Review #1**

**Rating:** 8

**Review:**

I think that the paper is very well written, I like it. The authors localized a phenomenon  and demonstrated how to exploit it. I trust the results because I performed exactly the same experiments for CIFAR-10 with longer non-regularization periods and found that there is no effect (this is also that the authors show in the paper)  but I didn't test on other datasets and obviously didn't think about potential benefits for compression.
Since model compression is not my field, I would just make a general suggestion to include and compare to state-of-the-art techniques.

---
Update:
The authors updated the paper and replied to all reviewers in detail. I still think that the paper should be accepted.

**Experience Assessment:**

I have published one or two papers in this area.

**Review Assessment: Checking Correctness Of Derivations And Theory:**

I assessed the sensibility of the derivations and theory.

**Review Assessment: Checking Correctness Of Experiments:**

I assessed the sensibility of the experiments.

**Review Assessment: Thoroughness In Paper Reading:**

I read the paper thoroughly.

---

> ### Author Response · Authors · 2019-11-11
> **Response to AnonReviewer1**
>
> We would like to thank you for the review and comments.
>
> We are very happy to hear that you also experienced the same experiments for CIFAR-10. Even though regularization with different pNR is not a main topic in this paper, model accuracy is improved by using large pNR as shown in Figure 11 with large learning rates. Extending our work to be focused on regularization using recently suggest regularization techniques would be an interesting future research topic.
>
> In Section 5, we added a few related works with the key words of 'periodic compression' while such techniques are also applicable to our work.

---

### Official Review · AnonReviewer4 · 2019-11-02
**Official Blind Review #4**

**Rating:** 3

**Review:**

This work investigates compression-aware training and introduces a new hyper-parameter called "Non-regularization period". Basically, the paper proposes to apply weight regularization and compression less frequently so that they can use stronger regularization/compression, hence achieving high compression ratio and model accuracy.

Overall, the idea of asynchronous regularization and compression is interesting and worth further investigation. However, I vote for a reject for now because
(1) The paper is hard to follow and the writing can be significantly improved especially for the abstract and introduction. I felt rather confused when I first read the abstract and introduction. The authors frequently use the word "weight update", however it is unclear whether it refers to gradient update or the update caused by regularization/compression. Besides, the title is also a little confusing. There's little discussion about the choice of batch size and I don't really understand how weight regularization is decoupled from batch size after reading the paper. I suggest the authors to think more about the title.
(2) The reason why such a training scheme improves model compression is unclear and needs further investigations. In the paper, the authors first interpret model compression as a way of inducing weight decay (and random weight noise). Particularly, the model accuracy is roughly constant over different value of NR period (as shown in Figure 3 and Figure 4) when weight decay is used. However, the results of model compression are quite different (according to Figure 5). The best performance is only achieved with very large NR period in the case of SVD and pruning. I think the authors should give some explanations for that difference. Also, I notice that quantization behaves more similarly to weight decay and it requires further discussion.

To sum up, I like the idea of asynchronous regularization/compression, but I'm not quite satisfied with current version of paper. I encourage the authors to improve their writing and add more discussions (see my point (2)) to the paper. I'm willing to increase my score if the authors can address my concerns.

**Experience Assessment:**

I have published one or two papers in this area.

**Review Assessment: Checking Correctness Of Derivations And Theory:**

I assessed the sensibility of the derivations and theory.

**Review Assessment: Checking Correctness Of Experiments:**

I assessed the sensibility of the experiments.

**Review Assessment: Thoroughness In Paper Reading:**

I read the paper at least twice and used my best judgement in assessing the paper.

---

> ### Author Response · Authors · 2019-11-11
> **Response to AnonReviewer4**
>
> We would like to thank you for the review and comments.
> We revised the manuscript and added discussions and experimental results to address your concerns.
>
> 1) We agree that "weight update" might be confusing to the readers. In the revised manuscript, for every weight manipulation to compress weights, we chose "weight regularization" to be distinguished from weight updates for gradient descent, not only in abstract and introduction, but also in the entire manuscript. For batch size selection, we revised introduction and Section 5 to clarify that batch size is selected to be small for better generalization while weight regularization does not have to be performed for every mini-batch. We would be very happy if you can suggest any other parts to be improved in writing. Unfortunately, we cannot edit the title at this moment. We will find a chance to revise the title appropriately for the final manuscript.
>
> 2) We acknowledge that Figure 5 may look different from Figure 3 and Figure 4 without detailed discussions, especially for SVD and pruning. To explain how those figures are closely related, we added experimental results (Figure 6) and detailed discussions in Section 5 while Figure 5 and 14 include additional results (using wider hyper-parameter exploration). In Section 4, the key message is that the amount of weight regularization divided by pNR is almost constant (hence, longer pNR allows stronger weight regularization). Unlike weight decay, compression ratio is not directly related to the average amount of weight regularization. Hence, we first obtained weight difference by compression and found that there is a particular optimal constant (weight difference)/(pNR) value for the best model accuracy regardless of compression ratio. Since model compression involves much more complicated weight regularization effects than weight decay, Figure 5 can be different from Figure 4. But in Figure 6, we observed that achieving an optimal constant of (weight difference)/(pNR) can explain both Figure 4 and Figure 5.

---

> ### Comment · AnonReviewer1 · 2019-11-14
> **Title**
>
> Feel free to let us know if the updated paper addresses some of your concerns, the paper is currently borderline.

---

### Author Response · Authors · 2019-11-14
**Revised Manuscript is uploaded**

Revised Manuscript Uploaded

To address concerns and answer questions of the reviewers, we revised manuscript with the following major changes:

- Figure 5 includes additional experimental results with extended hyper-parameter explorations compared with the original manuscript (Figure 14 in Appendix shows test perplexity for each experiment.
- Figure 6 is added to explain how Figure 5 (relationship between pNR and compression ratio) is aligned with Figure 3 or Figure 4 along with detailed discussions in Section 5.
- We mainly revised abstract, Section 1, and Section 5.

- To avoid any confusion of the usage of "weight update", we explicitly used the terms "weight regularization for compression" whenever weights are manipulated for model compression in the entire manuscript.
- We added the discussion of the batch size choice in abstract and introduction to clarify that pNR should be decoupled from batch size (please refer to Section 5)
- We replaced 'asynchronous regularization' with 'occasional regularization' to address the concern of Reviewer 2.
- We tried to clarify the relationship between regularization and model compression.
- We added more detailed discussions in Section 4 and 5 to explain the Figure 4 and 5.

---

### Decision · Program_Chairs · 2019-12-19

**Decision:**

Reject

**Comment:**

This paper proposes to apply regularizers such as weight decay or weight noise only periodically, rather than every epoch. It investigates how the "non-regularization period", or period between regularization steps, interacts with other hyperparameters.

Overall, the writing feels somewhat scattered, and it is hard to identify a clear argument for why the NRP should help. Certainly one could save computation this way, but regularizers like weight decay or weight noise incur only a small computational cost anyway. One explicit claim from the paper is that a higher NRP allows larger regularization. There's a sense in which this is demonstrated, though not a very interesting sense: Figure 4 shows that the weight decay strength should be adjusted proportionally to the NRP. But varying the parameters in this way simply results in an unbiased (but noisier) estimate of gradients of exactly the same regularization penalty, so I don't think there's much surprising here.

Similarly, Section 3 argues that a higher NRP allows for larger stochastic perturbations, which makes it easier to escape local optima. But this isn't demonstrated experimentally, nor does it seem obvious that stochasticity will help find a better local optimum.

Overall, I think this paper needs substantial cleanup before it's ready to be published at a venue such as ICLR.